# How Diffusion Impacts Cortical Protein Distribution in Yeasts

**DOI:** 10.3390/cells9051113

**Published:** 2020-04-30

**Authors:** Kyle D. Moran, Daniel J. Lew

**Affiliations:** Department of Pharmacology and Cancer Biology, Duke University Medical Center, Durham, NC 27710, USA; kyle.moran@duke.edu

**Keywords:** diffusion, cell polarity, Cdc42

## Abstract

Proteins associated with the yeast plasma membrane often accumulate asymmetrically within the plane of the membrane. Asymmetric accumulation is thought to underlie diverse processes, including polarized growth, stress sensing, and aging. Here, we review our evolving understanding of how cells achieve asymmetric distributions of membrane proteins despite the anticipated dissipative effects of diffusion, and highlight recent findings suggesting that differential diffusion is exploited to create, rather than dissipate, asymmetry. We also highlight open questions about diffusion in yeast plasma membranes that remain unsolved.

## 1. Introduction

The growth mode of budding yeasts, including the popular model *Saccharomyces cerevisiae*, involves the creation of a new cell (the bud) connected to a mother cell. Bud formation involves sequential changes in the pattern of secretion during the cell cycle, changing from uniform secretion in the mother during G1 phase, to polarized secretion into the bud during S/G2/M phases, to directed secretion towards the mother-bud neck during septation (Figure 1A). This pattern of secretion means that new proteins are delivered to locations that vary depending on when in the cell cycle the proteins are synthesized. For proteins that are linked to the cell wall, where they do not subsequently diffuse, the timing of synthesis determines their spatial distributions, as shown by elegant promoter-swap experiments [1] (Figure 1B). Similar analyses indicated that some “landmark” proteins, which guide the positioning of future bud sites, are also localized by controlling the timing of their expression in the cell cycle [2]. However, it was long assumed that the dissipative effects of diffusion would make this mechanism ineffective for proteins that are not attached to the rigid cell wall.

## 2. Slow Diffusion Restricts Mobility in the Yeast Plasma Membrane

Yeast cells are small, with diameters of 4–6 µm [3]. Over such distances, proteins with diffusion constants typical of most integral membrane proteins in animal and bacterial membranes (~0.1 µm^2^/s) [4,5,6,7] would diffuse across the cell in a few minutes. Nevertheless, many yeast membrane proteins accumulate asymmetrically in the mother or bud plasma membrane. Early hypotheses to explain such asymmetries involved a diffusion barrier at the mother-bud neck linked to cytoskeletal septin filaments found at that location [8,9]. However, subsequent work suggested instead that no barrier is needed because diffusion in the yeast plasma membrane is remarkably slow [10].

Fluorescence Recovery After Photobleaching (FRAP) studies showed that single-pass transmembrane proteins in the yeast plasma membrane diffuse with D < 0.0025 µm^2^/s, while multipass transmembrane proteins diffuse with D < 0.0005 µm^2^/s [11,12]. Because these studies assumed that all fluorescence recovery was due to diffusion (neglecting other possible recovery pathways), they represent upper bounds on the diffusion constant, so these findings indicate that proteins in the yeast plasma membrane are remarkably immobile. This does not appear to stem from unusual features of the proteins themselves, as similar proteins targeted to a yeast vacuolar membrane or a mammalian plasma membrane are far more mobile (D ~0.1 µm^2^/s) [11]. Consistent with the cited studies, FRAP experiments on a variety of other membrane proteins detected very little recovery on 10 min timescales, in fission yeast as well as budding yeast [13,14,15]. Thus, it appears that yeast plasma membranes severely restrict diffusion of embedded proteins.

## 3. Basis for Asymmetric Distribution of Integral Membrane Proteins

Interestingly, several long-lived membrane proteins are enriched in the mother plasma membrane, and this has been linked to the process of aging in yeast [13,16,17]. The proton pump, Pma1, is enriched in the mother and is thought to initiate aging by raising the cytoplasmic pH in the mother cell [16]. Transporters of the multidrug resistance family are also enriched in the mother plasma membrane, and are thought to decay with age, depriving the aging mother cell of detoxification capacity [13].

Given the very restricted lateral mobility of plasma membrane proteins in yeast, one way to obtain mother-enriched protein localization would be to restrict protein synthesis to a specific phase of the cell cycle, as discussed for cell wall proteins (Figure 1B). Indeed, the timing of synthesis combined with slow diffusion appears to account for much of the observed asymmetry for these long-lived proteins [10].

Slow diffusion also has implications for the distribution of short-lived proteins. The yeast pheromone (α-factor) receptor, Ste2, undergoes endocytosis and vacuolar degradation [18,19]. In cells that have not been exposed to pheromone, Ste2 synthesis is constitutive and endocytosis is slow. This causes Ste2 to accumulate in broad zones representing sites of recent delivery that change through the cell cycle [12,20]. In cells exposed to pheromone, Ste2 endocytosis and degradation is rapid, so that only very recently deposited receptor is present on the plasma membrane, accumulating more tightly around the site of polarized secretion [21].

Endocytosis and recycling can also change the distribution of long-lived membrane proteins. In a series of elegant experiments, the polarized distribution of v-SNAREs on the yeast plasma membrane was shown to arise from the combination of polarized delivery, slow diffusion, and rapid endocytosis that recycles v-SNAREs before they diffuse too far from their site of deposition [11] (Figure 2A). Similar mechanisms account for the polarized distribution of the plasma membrane stress sensor Wsc1 [22]. Point mutations that disable the endocytosis of these proteins cause them to accumulate uniformly all over the membrane, while appending a strong endocytosis signal on an otherwise uniformly distributed plasma membrane protein suffices to make its distribution resemble that of v-SNAREs and stress sensors [11].

In aggregate, these studies highlight two key aspects of yeast cell biology that enable asymmetric accumulation of integral membrane proteins. First, secretion is directed towards different sites in a stereotypical manner during the cell cycle, allowing targeted delivery of proteins depending on their time of synthesis. Second, slow diffusion in the yeast plasma membrane can allow for asymmetries to persist, either passively for long-lived almost immobile multipass membrane proteins, or via endocytic recycling for more mobile single-pass membrane proteins.

## 4. Asymmetric Distribution of Peripheral Membrane Proteins: Recycling via the Cytoplasm

Unlike integral membrane proteins, peripheral membrane proteins can detach from the membrane to the cytoplasm, where diffusion is much more rapid (typically 5–15 µm^2^/s) [6,7]. For a cell as small as yeast, cytoplasmic diffusion is very effective at dissipating any spatial gradients, as proteins would diffuse throughout the cell in ~1 s. However, selective attachment/detachment from membranes can result in localization to discrete sites. Perhaps the most intensively studied example is the conserved Rho-family GTPase Cdc42, responsible for polarity establishment.

Cdc42 localization changes dramatically through the cell cycle (Figure 2B). Initially dispersed throughout the cell, in late G1 Cdc42 accumulates at a patch on the plasma membrane that defines the site of future bud emergence. Cdc42 is enriched in the bud membrane during bud growth and relocates to the mother-bud neck during cytokinesis [23,24]. Because Cdc42 organizes the downstream polarization of the cytoskeleton, considerable interest has focused on understanding how Cdc42 itself becomes enriched at the polarity site [25].

Cdc42 is prenylated, which promotes its attachment to membranes. An early hypothesis to explain Cdc42 polarization was that, as for the v-SNAREs discussed above, directed secretion combined with endocytosis could mediate asymmetric accumulation of Cdc42 [26] (Figure 2A). However, estimates of the Cdc42 diffusion constant (0.036 µm^2^/s: [27]) were far higher than those for v-SNAREs (0.0025 µm^2^/s: [11]). With such high mobility, no realistic endocytic pathway could recycle the Cdc42 fast enough to counteract the dissipative effect of diffusion [28,29]. A proposed role for the septin diffusion barrier in maintaining Cdc42 polarity [30] also seems unlikely, as Cdc42 remains well polarized in septin mutants. Instead, current models of Cdc42 localization invoke recycling through detachment of the protein to the cytoplasm.

At the membrane, Cdc42 switches between GTP- and GDP-bound states in a manner assisted by GDP exchange factors (GEFs) and GTPase-activating proteins (GAPs) (Figure 2C). The Cdc42-directed GEF in yeast is primarily cytoplasmic, but associates with other proteins that allow it to be recruited to membrane sites with pre-existing GTP-Cdc42 [31]. This means that locations with some GTP-Cdc42 accumulate GEF, which loads GTP on neighboring Cdc42, creating a positive feedback loop that can drive symmetry breaking polarization. As GTP-Cdc42 diffuses, ubiquitous GAPs promote its conversion to GDP-Cdc42. GDP-Cdc42 can detach from the membrane to the cytoplasm, assisted by a GDP-dissociation inhibitor (GDI) [32] (Figure 2C) or via other poorly characterized pathways [33]. This enables rapid dissipation of any GDP-Cdc42 asymmetry via cytoplasmic diffusion. The preferential detachment of GDP-Cdc42 (compared to GTP-Cdc42) enables recycling of GDP-Cdc42 via the cytoplasm while maintaining a gradient of GTP-Cdc42 at the membrane (Figure 2C) [34,35,36]. This mechanism resembles the endocytic recycling mechanism of v-SNAREs (Figure 2A), but with rapid cytoplasmic diffusion allowing for much faster recycling that maintains the asymmetry of Cdc42 despite the higher diffusion of this peripheral membrane protein. In addition to cytoplasmic recycling, some studies have suggested that Cdc42 polarization is assisted by differential diffusion of Cdc42 at the plasma membrane.

## 5. Differential Diffusion of Cdc42 at the Membrane

The first report to consider differential diffusion of Cdc42 suggested that Cdc42 could diffuse more slowly in one part of the plasma membrane than another, perhaps due to membrane domains of different lipid composition [37]. Such location-dependent diffusion would enable diffusion to promote, rather than dissipate, Cdc42 asymmetry at the membrane. The notion that differential diffusion would serve to generate a nonhomogeneous distribution is widely recognized in physics and chemistry. As a simplifying case, let us neglect membrane–cytoplasm exchange and consider a cell that starts with uniformly localized Cdc42 all over the plasma membrane. If Cdc42 diffusion were reduced in a specific zone of the membrane, then molecules entering that zone by rapid lateral diffusion would slow down, and Cdc42 would accumulate in the slow-diffusion zone (Figure 3A). At steady state, Cdc42 concentration would be uniformly high within the zone, and uniformly low outside the zone. At the zone boundary, there would be a balance between escape of the slow-moving Cdc42 and arrival of rapidly-diffusing Cdc42. To maintain that balance, the concentration of Cdc42 within the slow zone must be higher than that outside the zone, by a factor equal to the ratio of diffusion constants.

Evidence for such location-dependent diffusion came from an unexpected observation [37]. GFP-Cdc42 at polarity sites displayed patchy localization, with dark spots in an otherwise bright polarized zone (Figure 3B). The patchiness persisted in mutants lacking the GDI and treated with Latrunculin to disable endocytosis: conditions intended to disable membrane–cytoplasm exchange. Moreover, an iFRAP experiment bleaching all of the GFP-Cdc42 outside of the polarity site led to a gradual loss of fluorescence over 2–4 min while maintaining the patchy pattern at the membrane. How could steep local gradients in GFP-Cdc42 concentration between adjacent patches exist when diffusion should rapidly dissipate such gradients?

The authors proposed that the patches represented zones in which Cdc42 mobility was altered, so that bright patches were slow-mobility zones and dark patches (as well as areas outside the polarity site) were high-mobility zones. In this view, location-dependent diffusion explained the GFP-Cdc42 localization pattern (Figure 3C). Fitting the GFP-Cdc42 localization data yielded diffusion estimates of 0.053 µm^2^/s (fast zones) and 0.0061 µm^2^/s (slow zones). The authors speculated that fast and slow diffusion zones might represent patches of membrane with different lipid composition, reminiscent of lipid rafts, although current models for lipid rafts consider much smaller and more transient entities.

An alternative, and in our view more likely, interpretation of the bright and dark patches is based on the observation that fluorescent markers of endocytic sites at the membrane showed a pattern that was anticorrelated with the GFP-Cdc42 signal [37]. We suggest that this could arise if GFP-Cdc42 were excluded from endocytic sites by the clathrin-associated coat proteins, generating the dark patches in the polarity site (Figure 3D). Because Latrunculin treatment blocks internalization of the endocytic coat, endocytic “dark patches” could stably persist in the iFRAP experiment described above. This hypothesis would obviate the need for location-dependent diffusion to explain the existence of stable bright and dark patches. Instead, the gradual dissipation of GFP-Cdc42 could reflect maintenance of asymmetry by GDI-independent recycling via the cytoplasm [33], combating a uniform diffusion of Cdc42.

## 6. Differential Diffusion of GTP-Cdc42 and GDP-Cdc42

Other studies proposed a distinct scenario in which diffusion of any given Cdc42 molecule is similar at all locations, but different forms of Cdc42 (GTP-Cdc42 vs. GDP-Cdc42) have inherently different mobility [14,38]. Consider a situation in which GDP-Cdc42 diffuses more rapidly in the plane of the membrane than GTP-Cdc42 (Figure 3E). The positive feedback loop mentioned above (Figure 2C) would create a zone where GEF activity is elevated, so that most Cdc42 in that zone would be GTP-bound. Assuming that ubiquitous GAP activity keeps Cdc42 elsewhere mostly GDP-bound, the cell would develop a zone with slow (GTP-)Cdc42 diffusion surrounded by a membrane with rapid (GDP-)Cdc42 diffusion. As with the differential mobility that comes from preferential detachment of GDP-Cdc42 from the membrane to the cytoplasm, this version of differential diffusion enables (rather than dissipating) polarization, and can collaborate with positive feedback via GEF recruitment to concentrate Cdc42 at the polarity site.

Convincing evidence for a stark GDP/GTP-dependent difference in Cdc42 diffusion came from studies in fission yeast, where (unlike in budding yeast) Cdc42 mobility was largely independent of the GDI. GDP-Cdc42 was estimated to diffuse rapidly in the membrane (0.1-0.2 µm^2^/s), while GTP-Cdc42 diffusion was at least 10-fold slower [14]. This difference is sufficient, when combined with local GEF activity, to explain the enrichment of Cdc42 at polarity sites. Although early estimates in budding yeast found no evidence for a difference in diffusion constants for GDP-Cdc42 and GTP-Cdc42 [27], more recent work using single-particle tracking of photo-activatable mEOS-Cdc42 is consistent with an approximately two-fold difference (0.011 µm^2^/s for GTP-Cdc42 and 0.023 µm^2^/s for GDP-Cdc42) [38]. These findings suggest that both yeasts exploit differential diffusion of GDP-Cdc42 and GTP-Cdc42 to enhance Cdc42 asymmetric distribution at the membrane.

## 7. Conclusions and Open Questions

Diffusion is best known as a dissipative process that acts to reduce concentration gradients in cells and membranes. Nevertheless, many proteins in yeast plasma membranes develop markedly asymmetric patterns of accumulation. The studies discussed above suggest that this is enabled by two features. First, protein diffusion in the yeast plasma membrane is unusually slow, which reduces the dissipative effects of diffusion. Second, some proteins, including Cdc42, may diffuse at different rates depending on their conformation. These studies raise several unsolved questions, a few of which are highlighted below.

### 7.1. Why Is Diffusion So Slow in the Yeast Plasma Membrane?

Mutations that perturb the normal lipid composition of the plasma membrane can modestly increase protein diffusion [11,15]. Thus, an unusual lipid composition may increase membrane viscosity, hence slowing diffusion. However, even quite dramatic changes in lipid composition do not restore diffusion to the levels typical of other membranes, suggesting that non-lipid factors also contribute to immobilization. Because the plasma membrane is in contact with the rigid cell wall, it seems likely that interactions between membrane proteins and the immobile wall may also serve to restrict lateral mobility. Studies in plant cells have shown that even membrane proteins that do not bind to the cell wall can be corralled due to turgor-based apposition of the plasma membrane to the wall [39]. In yeast there are other immobile structures called eisosomes that form stable grooves in the plasma membrane and can impede diffusion of other membrane proteins [40]. Moreover, abundant membrane proteins like the proton pump Pma1 display dynamic mesh-like localization patterns in the membrane [41,42], potentially creating corrals that could slow diffusion of other proteins by confining their movement [43]. Understanding the contributions made by lipids, proteins, and the cell wall would be interesting and informative with regard to the degree to which similar phenomena occur in other systems.

### 7.2. Why Do So Many Membrane Proteins Accumulate Asymmetrically?

With the exception of the bud-site selection landmarks [2] and perhaps the stress sensors [22], we do not understand why a cell would benefit by accumulating many membrane proteins asymmetrically. Indeed, in several cases it seems counterproductive to do so. Asymmetric accumulation of proton pumps and transporters has been linked to deterioration of mother cells during aging [13,16], and it is unclear what benefit might outweigh that cost. Asymmetric accumulation of pheromone receptors has the potential to mislead mating cells about the location of a mating partner, requiring corrective mechanisms to provide accurate detection [12]. It would seem quite simple to avoid these problems, and there are instances where endocytic recycling has been tuned so as to counteract the asymmetry derived from timed synthesis in the cell cycle [10]. Why, then, are apparently unproductive asymmetries maintained?

### 7.3. What Is the Basis for the Differential Diffusion of GDP-Cdc42 and GTP-Cdc42?

The structures of GDP- and GTP-bound Cdc42 differ in well-appreciated ways [44], but it is not clear why that conformational change would significantly alter the protein’s mobility in the membrane. Perhaps the simplest hypothesis would be that there is a much lower-mobility integral membrane protein that interacts selectively with GTP-Cdc42, slowing its diffusion. Single-particle tracking experiments in plant and animal as well as yeast cells identified a subset of “immobile” molecules, and high variability in the trajectories of individual molecules, consistent with interactions that transiently immobilize the protein. Other potential reasons for such transient immobilization invoke roles for lipids and nanoclusters of Cdc42 [38,45,46,47]. Overall, it seems we still have much to learn about how various proteins diffuse in the yeast plasma membrane.

## Figures and Tables

**Figure 1 cells-09-01113-f001:**
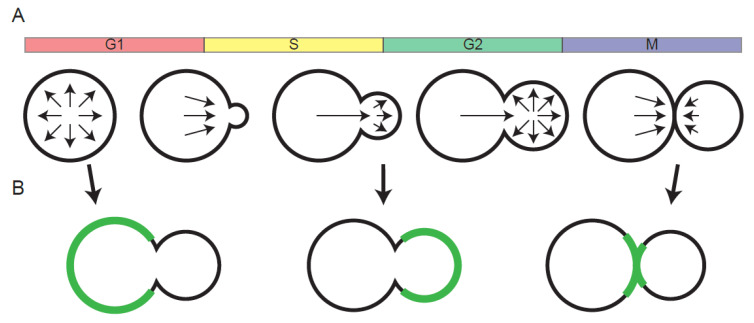
Stereotyped pattern of secretion enables localization dependent on the time of synthesis during the cell cycle. (**A**) Arrows indicate secretion patterns through the cell cycle. (**B**) With no diffusion, cell wall-linked proteins are spatially distributed in a manner that reflects where they were initially secreted. This leads to their accumulation in the mother, the bud, or at sites of cytokinesis, depending on the timing of their synthesis during the cell cycle.

**Figure 2 cells-09-01113-f002:**
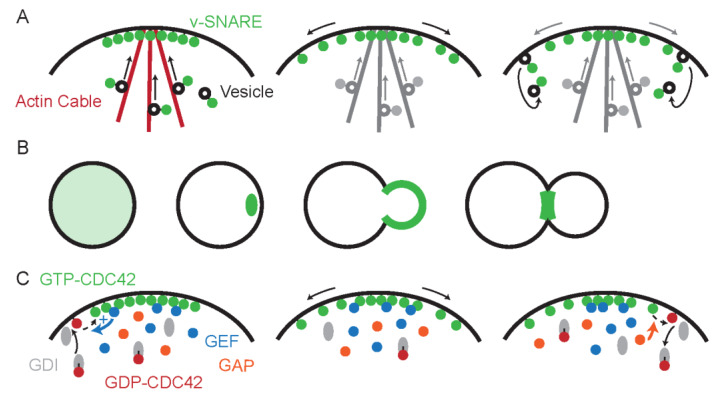
Dynamic localization of integral membrane and peripheral membrane proteins. (**A**) Asymmetric distribution of v-SNAREs arises from vesicle-mediated delivery of v-SNAREs to polarity sites (left), slow diffusion of v-SNAREs on the membrane (middle), and recycling via endocytosis (right). Note that while these processes are highlighted in separate panels for clarity, they all occur continuously. (**B**) Cdc42 localization through the cell cycle. (**C**) Asymmetric distribution of GTP-Cdc42 arises from deposition of GDP-Cdc42 on the membrane by GDI followed by local activation by GEF (left), diffusion of GTP-Cdc42 on the membrane (middle), and inactivation by GAP allowing GDI to pluck GDP-Cdc42 off the membrane, recycling it to the cytoplasm (right). Note that while these processes are highlighted in separate panels for clarity, they all occur continuously. Recruitment of GEF to sites with GTP-Cdc42 enables a positive feedback loop that loads GTP on neighboring Cdc42.

**Figure 3 cells-09-01113-f003:**
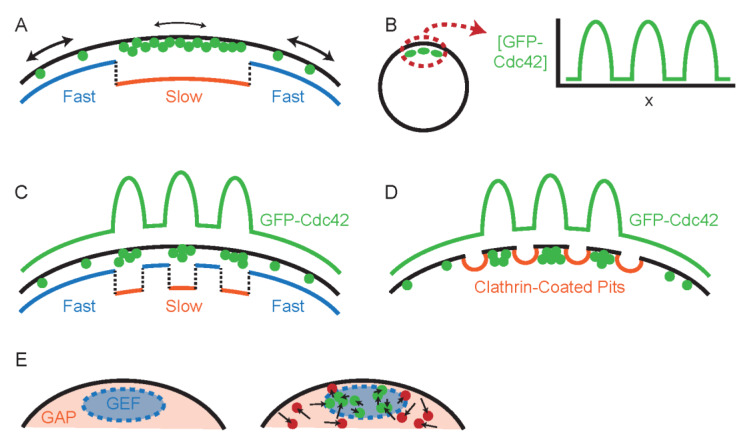
Differential diffusion of Cdc42. (**A**) Location-dependent differential diffusion: zones of fast and slow mobility (blue and orange, respectively) would cause a protein (green) to become concentrated in the slow-diffusion zone. (**B**) GFP-Cdc42 displays patchy localization at polarity sites. (**C**) Patchy localization could be explained by alternating zones of slow and fast mobility. (**D**) Patchy localization of GFP-Cdc42 could also be explained by exclusion of GFP-Cdc42 from endocytic sites (clathrin-coated pits). (**E**) Differential diffusion of GTP-Cdc42 and GDP-Cdc42. Localized GEF activity (blue) and uniform GAP activity (orange) would lead to local activation of Cdc42 in the GEF zone. If GTP-Cdc42 (green) diffuses less than GDP-Cdc42 (red), then Cdc42 accumulates in the GEF zone. Arrows indicate high mobility of GDP-Cdc42 relative to GTP-Cdc42.

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
