# Peer review of "How Diffusion Impacts Cortical Protein Distribution in Yeasts"

_cells, 2020, doi:10.3390/cells9051113_

Round 1

Reviewer 1 Report

How cells develop polarity with asymmetric distribution of membrane proteins is highly topic and of broad interest. This review describes several models that explain the formation of asymmetric distribution of membrane proteins in yeast. This paper particularly focuses on the differential diffusion mechanism. I think this is good review about what is known about the asymmetry formation in yeast, and is suitable for publication in this journal after the authors address the following several minor points.

1. In the paragraph starting in line 106, the authors explain the current model of the formation of Cdc42 asymmetric distribution. While the given mechanistic explanation seems to be sufficiently complex, how the overall scheme generates the asymmetry is not so clear to me. The authors should give an intuitive explanation of how the system evolves the asymmetry breaking a uniform state. A positive feedback seems to work for the asymmetry formation. This keyword appears only in the figure legend. The accompanying figure (Fig. 3c) contains the reaction schemes with some arrows (which are a bit small to see). But, from these arrows, it is difficult to see how the positive feedback loop works to form the asymmetry.

2. The authors explain how differential diffusion performs to generate a polarized state. If there is a zone in which diffusion constant is different from the surrounding area, such a difference can make nonuniform distribution in the membrane protein. But the next question naturally arises as to how such nonuniform zones are formed on the plasma membrane. There is no mention on this point. The authors should mention this next question.

3. For the case of Cdc42, if the diffusion constants of GTP-Cdc42 and GDP-Cdc42 are different, and if there is an active GEF zone, this leads to the formation of an asymmetric distribution of GTP-Cdc42. But what makes this zone of active GEF? The authors should mention this question. How compatible does this differential diffusion model with the positive feedback model introduced previously? Is there a possibility that both mechanisms work together for the formation of asymmetry distribution of Cdc42?

Author Response

1. We have added the “positive feedback” idea to the text (lines 112-113) and modified Fig. 2C to highlight this feature.

2. We have added text (lines 151-154) on the speculation offered by Slaughter et al. that different zones might have different lipid composition, but also added a comment that this does not fit well with current models of lipid raft size/longevity.

3. We envision that positive feedback is what creates the zone of active GEF, and have edited the text (lines 170-172) to clarify this point.

Reviewer 2 Report

Moran and Lew discuss how differential protein diffusion is used to create and maintain plasma membrane asymmetry in yeast. This opinion initially reviews how the rate of diffusion is combined other processes (timing of synthesis, or protein turnover by degradation/endocytosis, selective detachment) to control asymmetry of plasma membrane proteins and then focus on the generation of local asymmetries for the small Rho-family GTPase Cdc42. 

The manuscript is very well written, and nicely exposes open questions and current models that are clearly illustrated by the accompanying figures. I consider that it will be of great interest for yeast researchers, and also has broad interest for researchers studying cell asymmetry. I strongly support publication as it is, and I only have a minor comment that the authors may consider:

It is interesting that diffusion is particularly slow in the yeast plasma membrane, which may be linked to yeast-specific features (e.g. cell wall or eisosomes as the authors refer). To further emphasise that this is a particular feature of diffusion in the yeast plasma membrane, it may be relevant to refer to studies that measured diffusion constants of cortical/membrane proteins that display asymmetric distribution in animal cells.

Author Response

We now cite studies in both animal and bacterial membranes that found higher diffusion constants (line 33).

Reviewer 3 Report

This manuscript (ID cells-767411) for the Opinion article entitled "How diffusion impacts cortical protein distribution in yeasts" reviews how yeast cells achieve asymmetric distribution of membrane proteins focusing on the roles of diffusion in protein asymmetric distribution in yeasts. One of the author's major claim is apparently that, while the diffusion indeed works to dissipate it, it is also playing the opposing roles when the cell has differential diffusion.

The authors' logic and claims sounds reasonable. Their explanations are understandable. I think this manuscript needs only minor improvements to enrich the information for general readers. I have listed up several specific minor comments below:

Throughout the manuscript: The authors may add missing citations to support the facts given in some of their sentences, e.g.,

  • page 2 line 32: "typical of most integral membrane proteins in biological membranes (~0.1 um2/s)"
  • page 2 line 36: "subsequent work suggested instead that no barrier is needed because diffusion in the yeast plasma membrane is remarkably slow."
  • page 2 line 40: "most technical issues with diffusion estimates lead to overestimation of the diffusion constant (e.g. unappreciated recovery pathways that occur in parallel with diffusion),"
  • page 3 line 86: "to the cytoplasm, where diffusion is much more rapid (typically 5-15 um2/s)."
  • page 5 line 158: "Other studies proposed a distinct scenario in which diffusion of any given Cdc42 molecule is similar at all locations, but different forms of Cdc42 (GTP-Cdc42 vs GDP-Cdc42) have inherently different mobility."

Citing proper references may help the general readers (in particular the non-specialist readers) to follow the written facts.

Introduction: I wonder what is the advantage or motivation to focus on the yeast for this subject. The authors may explain it.

Conclusion: I also wonder whether these understandings for yeast can be extendable to the other cells' case. The authors might put examples showing roles of diffusion for asymmetric protein accumulation in other kinds of cells (either in Introduction or Conclusion).

Page 2 line 45: "as similar proteins targeted to a yeast vacuolar membrane or a mammalian plasma membrane are far more mobile" How mobile is those proteins typically? The authors may put evaluations of diffusion constants for some examples.

Page 3 line 87: "For a cell as small as yeast, cytoplasmic diffusion is very effective" The authors may put evaluations of some quantities (e.g. typical time scale for the proteins to spread around the cell) to demonstrate the readers why and how effective it is in a small cell like yeast.

Page 3 line 100: "diffusion"  should be "diffusion constant"

Page 4 line 121: The concept itself of inhomogeneous distribution induced by position-dependent diffusion or differential diffusion is widely recognized in physics and chemistry fields. If possible, it might be good to briefly comment on such facts.

All figures: Resolutions of all figures should be increased. Some objects like arrows or something are apparently hard to recognize.

Fig. 1A: The authors should describe what do the arrows mean in the figure caption.

Author Response

Page 2, Line 32: We added the requested citations (line 33).

Page 2, Line 36: We added the requested citation (line 37).

Page 2, Line 40: We have changed the text to specify why the numbers are upper bounds (lines 40-44).

Page 3, Line 86: We added the requested citations (line 88).

Page 5, Line 158: We added the requested citations (line 169).

Introduction: The slow diffusion of proteins in the yeast plasma membrane is anomalous in that animal, plant, and bacterial membranes all have faster diffusion. This is perhaps a yeast-specific issue that we wanted to highlight. It is true that the differential diffusion part applies much more generally, but here we highlight the evidence for yeast.

Conclusion: See our response above.

Page 2, Line 45: We added the mobilities from the cited reference (line 47).

Page 3, Line 87: We added this timescale (~1 s) (lines 89-90).

Page 3, Line 100: Changed as recommended (line 102).

Page 4, Line 121: We do not claim that these concepts are novel, only the specific applications.

All Figures: The figures included in the PDFs uploaded alongside the manuscript are high-resolution.

Figure 1A: We have added the requested description.